# MULTI-PHYSICS OPERATOR NETWORK FOR IN-CONTEXT LEARNING (m-PHOeNIX)

## ABSTRACT

We propose a multi-physics operator network for simultaneous and sequential learning of solution operators of multiple heterogeneous parametric partial differential equations. Existing neural operators are adept at learning the solution operator of only a single physical system, and adapting to new physical equations requires training a new surrogate model from scratch with physics-specific intensive hyperparameter tuning. The proposed multi-physics neural operator leverages the recent advancements in wavelet-based kernel integral-induced neural operator modeling and instantiates a memory-based ensembling strategy for projecting heterogeneous physical systems into a common shared feature space. The local channel-level ensembling is supported by context gates, which not only utilize the shared features to embed the features of multiple heterogeneous physical systems into the network parameters but also allow the multi-physics operator to learn new solution operators by transferring knowledge sequentially; this allows the proposed model to continually learn without forgetting. We illustrate the efficacy of our algorithm by simultaneously and sequentially learning six complex time-dependent solution operators of six physical systems. The inference results on the simultaneous and sequentially trained models depict the ability to infer previously seen physical systems without fine-tuning and catastrophic forgetting, indicating the characteristics of a foundation model. The framework also demonstrates the super-resolution property and generalization to out-of-distribution input conditions.

## 1 INTRODUCTION

Scientific Machine Learning (SciML) involves the development of Machine Learning (ML) algorithms for solving physical systems governed by complex partial differential equations (PDEs), and has emerged as a computationally efficient alternative to classical numerical techniques like finite element method (FEM) (Hughes, 2012), spectral element method (SEM) (Lord et al., 2014), and finite volume methods (FVM) (Moukalled et al., 2016). The seminal works in SciML using modern ML methods for solving high-dimensional PDEs include neural networks (NNs) (Han et al., 2017; 2018; Sun et al., 2020), constrained NNs (Sirignano & Spiliopoulos, 2018; Zhu et al., 2019), variational NNs (Yu et al., 2018), and physics-informed NNs (Raissi et al., 2019). However, these frameworks require retraining from scratch as and when the input conditions, i.e., initial/boundary conditions, system parameters, etc., change. One potential alternative is to employ transfer learning (Goswami et al., 2020; Chakraborty, 2021; Chen et al., 2021); however, this is only effective for small perturbations in the input. The solution to this problem involves approximating operators of PDEs by using recently developed Neural Operators (NOs) (Li et al., 2020a;b; Gupta et al., 2021; Lu et al., 2021; Wang et al., 2021; Tran et al., 2021; Li et al., 2022; Tripura & Chakraborty, 2023; Hao et al., 2023; Navaneeth et al., 2024; Raonic et al., 2024). However, NOs are physics-specific, and as the governing physics of the underlying system changes, one needs to retrain the NOs from scratch. Additionally, as the NO is trained on the new physics, it forgets the previously learned operators. To address this apparent gap, we here propose the **m**ulti-**ph**ysics **o**perator **n**etwork for **i**n-context learning (m-PhOeNIX), which treats the PDEs of physical systems as one task and learns solution operators of multiple heterogeneous parametric PDEs in a single model. Besides simultaneous learning, m-PhOeNIX can also sequentially learn the operators' of new physical systems by ensembling the shared features of previously acquired operators without catastrophic forgetting.

The idea of in-context learning has gained some traction in the Natural Language Processing (NLP) community. In-context learning in Natural Language Processing (NLP) refers to a model's ability to perform tasks by interpreting and leveraging information provided within a specific context rather than relying solely on prior training. This approach allows models such as GPT2 (Radford et al., 2019), GPT3 (Brown et al., 2020), CLIP (Radford et al., 2021), ALIGN (Jia et al., 2021), PALM (Chowdhery et al., 2022), and REALM (Guu et al., 2020) to dynamically adapt their behavior based on the immediate context provided in the input prompt. The existence of such a concept of learning on multiple PDEs in SciML can be found in Yang et al. (2023); McCabe et al. (2023); Herde et al. (2024); Rahman et al. (2024); Hao et al. (2024). At the core, these frameworks use transformer-based sequence modeling approaches for homogenization of the operators of multiple physical systems to simultaneously learn solution operators of differential equations (DEs) from prompted data. The proposed m-PhOeNIX is a departure from this idea; instead, we utilize Green's integral (Duffy, 2015) kernel-based formulation. Green's formalism provides a direct connection to DEs in mathematics, whose success is not limited to classical mechanics (Hartmann, 2012) but in conjunction with nonlinear activation functions also evident in integral kernel-based NOs (Li et al., 2020a; Gupta et al., 2021; Tripura & Chakraborty, 2023; Rafiq et al., 2022). Note that the proposed m-PhOeNIX is the first operator learning approach that allows both multi-task (simultaneous) learning and continual learning without catastrophic forgetting.

The ability to reason and combine already learned tasks is a hallmark of intelligence. This requires modularity to support distributed learning and combinatorial strategy to meaningfully combine the previously acquired knowledge (Thrun, 1998). In the m-PhOeNIX framework, we achieve the modularity and meaningful combining of knowledge by instantiating a distributed learning strategy motivated from (Wang et al., 2020; Veness et al., 2021). We introduce expert wavelet integral blocks by parameterizing an ensemble of local integral kernels to ensemble task-specific NOs at the local kernel level. This enables distributed learning of different features of the multi-physics operator. Context gates are introduced to direct the predictions from local kernels toward a common operator by meaningfully weighing the local kernels based on the PDE label and context information. Overall, this work makes the following contributions: (1) It instantiates local kernel-level combinatorial representation learning strategy for the NOs. (2) It projects the physics of multiple heterogeneous systems into a common distributed feature space and simultaneously learns the multiple operators. (3) It performs combinatorial transfer of old operators to learn operators of new physical systems without catastrophic forgetting. We showcase the efficacy of m-PhOeNIX by simultaneously and sequentially learning operators of complex mechanics-oriented partial differential equations.

## 2 EXISTING WORKS

**Neural Operator.** Neural operators learn the discretization invariant solution operator of parametric PDEs, which are defined as a family of PDEs where the input conditions, such as the initial and boundary conditions (ICs/BCs), system parameters, source functions, etc., are allowed to vary over a finite range. Neural operators are trained only once, and once trained, the solutions for a new set of inputs require only a forward pass of the network. The literature on neural operators includes the universal approximation theorem-based deep operator network (DeepONet) (Lu et al., 2021), integral-kernel-based architectures like Fourier neural operator (FNO) (Li et al., 2020a), factorized Fourier neural operators (F-FNO) (Tran et al., 2021), wavelet neural operator (Tripura & Chakraborty, 2023), multiwavelet transform operator (MWT) (Gupta et al., 2021), spectral neural operator (SNO) (Fanaskov & Oseledets, 2023), spatio-spectral neural operator (SSNO) (Rafiq et al., 2022), and Laplace neural operator (LNO) (Cao et al., 2023), attention-based architectures like operator-former (Li et al., 2022), Gnot (Hao et al., 2023), waveformer (Navaneeth & Chakraborty, 2024), and convolution-based convolution neural operator (CNO) (Raonic et al., 2024). These architectures provide sufficiently accurate approximations to the solution operators of only one physical system or equations, and for every new physical system, a new neural operator has to be trained. Addressing this limitation is one of the primary concerns of this paper.

**Integral kernel based neural Operator.** Given the variable input conditions and solution spaces $\mathcal{A} := \mathcal{C}(\Omega; \mathbb{R}^{d_a})$ and $\mathcal{U} := \mathcal{C}(\Omega; \mathbb{R}^{d_u})$, where $\Omega \subset \mathbb{R}^d$ is a non-empty, bounded closed set denoting the solution domain, the neural operators approximate the solution operator $\mathcal{D} : \mathcal{A} \times \boldsymbol{\theta} \mapsto \mathcal{U}$ by parameterizing $\mathcal{D}$ with a finite parametric space $\boldsymbol{\theta}$ such an input $\boldsymbol{a} \in \mathcal{A}$ will be mapped to a unique

solution $\boldsymbol{u} \in \mathcal{U}$. Approximating the mapping $\boldsymbol{u}(x) = \mathcal{D}(\boldsymbol{a})(x)$ involves the following deep network, $\boldsymbol{u}(x) = \left( Q \circ q_h \circ \ldots \circ q_j \circ \ldots \circ q_0 \circ P \right)(\boldsymbol{a})(x)$, where $P : \mathbb{R}^{d_a} \mapsto \mathbb{R}^{d_v}$ increases the kernel dimension and $Q : \mathbb{R}^{d_v} \mapsto \mathbb{R}^{d_u}$ projects the feature space to solution space. The input $\boldsymbol{a}$ is first uplifted to $\boldsymbol{v}_0 = P(\boldsymbol{a})$, over which a series of iterative updates $q : \mathbb{R}^{d_v} \ni \boldsymbol{v}_j \mapsto \boldsymbol{v}_{j+1} \in \mathbb{R}^{d_v}$ are applied. The iterative updates $q$ are expressed using integral kernels as (Li et al., 2020a), $q_j(\cdot) := \Gamma\left( \mathcal{K}(\boldsymbol{a}; \phi \in \boldsymbol{\theta}) + g(\varphi \in \boldsymbol{\theta}) \right)(\cdot)$ for $j \in h$, where $\Gamma : \mathbb{R} \to \mathbb{R}$ is a point-wise nonlinear activation operator, and $\mathcal{K} \in \mathcal{C}(\Omega; \mathbb{R}^{d_v})$ is the integral operator. The integral operator $\mathcal{K}(\boldsymbol{a}(x); \phi)$ is defined as a convolution between the network kernel $k_\phi$ and input $\boldsymbol{v}_j(x)$ as, $\left( \mathcal{K}_\phi \boldsymbol{v}_j \right)(x) = \int_\Omega k_\phi\left( x - \xi \right) \boldsymbol{v}_j(\xi) d\xi$. The pointwise linear transformation $g : \mathbb{R}^{d_v} \to \mathbb{R}^{d_v}$ is modeled as a linear network or $1 \times 1$ convolution.

**Wavelet neural operator.** The method of performing the convolution $\left( \mathcal{K}_\phi \boldsymbol{v}_j \right)(x)$ varies across different architectures. In wavelet neural operator (WNO), the convolution is performed in the wavelet domain by projecting the inputs using wavelet decomposition. Given the wavelet and inverse wavelet transforms $\mathcal{W}$ and $\mathcal{W}^{-1}$, the parameterization of the kernel $k_\phi$ in the wavelet domain can be expressed as, $\left( \mathcal{K}_\phi \boldsymbol{v}_j \right)(x) = \mathcal{W}_\psi^{-1}\left( \mathcal{R}_\phi \cdot \left[ \mathcal{W}_\psi \boldsymbol{v}_j \right](s, r) \right)(x)$ where $s \in \mathbb{Z}^+$ and $r \in \mathbb{Z}$ denote the scale and translation parameters of the wavelet basis $\psi(x) \in L^2(\mathbb{R}^n)$. It is imperative to note that the kernels $\mathcal{R}_\phi$ are directly defined in the wavelet domain. Given the input $\boldsymbol{a} \in \mathbb{R}^{d \times d_a}$ in a domain with $d$ point discretization one has $\left[ \mathcal{W}_\psi \boldsymbol{v}_j \right](s, r) \in \mathbb{R}^{d_w \times d_v}$. By defining a network kernel $\mathcal{R}_\phi \in \mathbb{R}^{d_w \times d_v \times d_v}$, the kernel convolution is expressed as, $\left( \mathcal{R}_\phi \cdot \left[ \mathcal{W}_\psi \boldsymbol{v}_j \right](s, r) \right)_{jk} = \sum_{i=0}^{d_v} (\mathcal{R}_\phi)_{ijk} \left[ \mathcal{W}_\psi \boldsymbol{v}_j \right]_{ik}(s, r)$. However, the wavelet decomposition $\left[ \mathcal{W}_\psi \boldsymbol{v}_j \right](s, r)$ yields approximate ($\mathbb{A}_{\boldsymbol{v}}$) and detailed ($\mathbb{A}_{\boldsymbol{v}}$) wavelet coefficients of the input. The implementation of WNO, therefore, involves two kernel convolutions using the kernels $\mathcal{R}_\phi^{\mathbb{A}}$ and $\mathcal{R}_\phi^{\mathbb{D}}$. The kernel $\mathcal{R}_\phi^{\mathbb{A}}$ learns the features in the approximate space of the wavelet coefficients, and the kernel $\mathcal{R}_\phi^{\mathbb{D}}$ learns the features in the detailed space of wavelet coefficients.

**Multi-Physics operator.** The success of massive language models has recently spurred the development of large transformer-based multi-physics models for surrogate modeling in SciML. Among the primary multi-physics models are the transformer-based ICON (Yang et al., 2023) and MPP (McCabe et al., 2023). While ICON supports super-resolution features, it lacks the ability to fine-tune on downstream tasks. Conversely, MPP supports fine-tuning on downstream tasks but lacks superresolution properties. Recent transformer-based frameworks such as Poseidon (Herde et al., 2024), CoDA-NO (Rahman et al., 2024), and DPoT (Hao et al., 2024) tackle the challenges of both superresolution and fine-tuning on new PDEs. In these frameworks, for every new downstream task, a downstream model is trained by initializing the weights from the pre-trained model. However, these models do not support sequential learning, as each time a new downstream model is created, the previously learned features are often forgotten—leading to catastrophic forgetting of pre-trained tasks. To address this limitation, we propose a local ensembling strategy for pre-training and sequential learning of PDE operators. We focus on fine-tuning only a small part of the pre-trained network to sequentially adapt to downstream tasks, where we leverage newly learned features to sequentially learn new PDEs. We consider ICON and MPP as two representative frameworks of transformer-based multi-physics architectures for comparing our method.

## 3 Algorithm of m-PhOeNIX

The central idea in the m-PhOeNIX framework involves distributed operator learning of a diverse set of heterogeneous physical systems by projecting the set of systems into a common distributed space and later sequentially adapting the multi-physics operator to new physical systems without retraining the complete model. To address this, we introduce local wavelet experts to learn distinct features of heterogeneous physical systems. Secondly, we introduce the context gates to weight the local experts based on the task query, thereby supporting the transfer of knowledge during sequential learning and differentiating between tasks during prediction on previously learned systems.

**Local wavelet experts.** We define the local wavelet experts to be the wavelet convolution integrals used in the WNO architecture. Unlike in WNO, where the same wavelet basis is adopted across all the integral layers, we parameterize each local wavelet expert using a unique wavelet

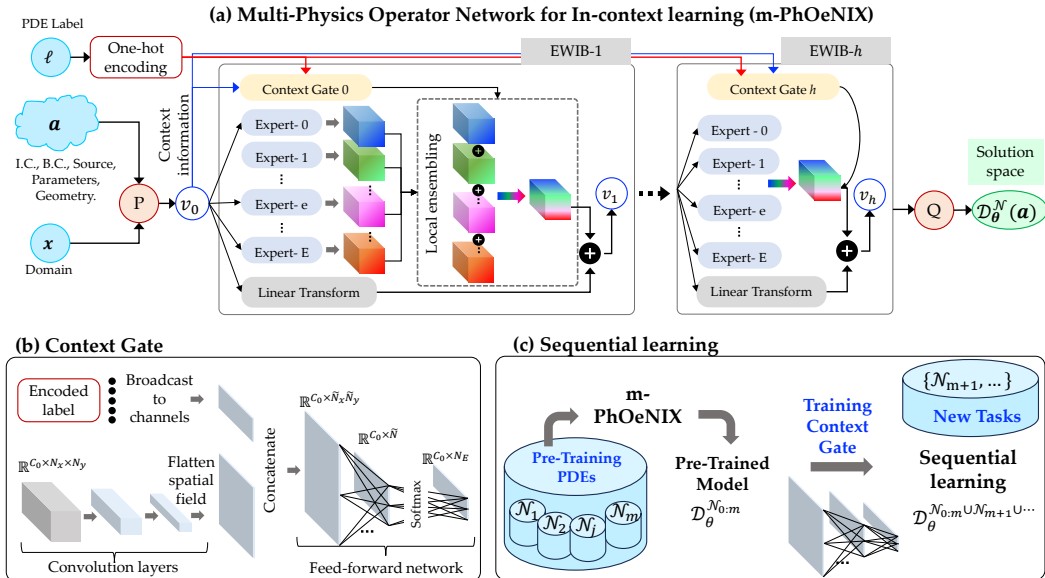

Figure 1: Architecture of m-PhOeNIX multi-physics operator learner. (a) The input $\{\boldsymbol{a}, x\}$ are encoded into $\boldsymbol{v}_0$ using P which is updated using EWIBs and later projected to solution $\boldsymbol{u}$ using Q. The ensembling between the local wavelet experts, indicated by the color gradients, is performed using the context gates. (b) The context gate uses the task label and context information to estimate the expert weights using an FFN. For 2D applications, the field size is first reduced using shallow CNN. (c) A m-PhOeNIX model is first pre-trained on a diverse set of pre-training physical systems. For new equations/operators, the pre-trained model is sequentially fine-tuned by backpropagating loss through the context gate parameters while fixing the m-PhOeNIX weights.

basis function $\psi_e$ for $e \in d_e$, where $d_e$ denotes the number of local experts inside an expert wavelet integral block to localize different feature of input in a separate wavelet space. The local convolution integrals involved in the parameterization of local wavelet experts are denoted as, $(\mathcal{K}_\phi^e \boldsymbol{v}_j)(x) = \int_\Omega k_\phi^e(x - \xi) \boldsymbol{v}_j(\xi) d\xi$, which in the wavelet domain is evaluated as, $(\mathcal{K}_\phi^e \boldsymbol{v}_j)(x) = \mathcal{W}_{\psi_e}^{-1}(\mathcal{R}_\phi^e \cdot [\mathcal{W}_{\psi_e} \boldsymbol{v}_j](s, r))(x)$. This yields the following kernel convolution, $(\mathcal{R}_\phi^e \cdot [\mathcal{W}_{\psi_e} \boldsymbol{v}_j](s, r))_{jk} = \sum_{i=0}^{d_v} (\mathcal{R}_\phi^e)_{ijk} [\mathcal{W}_{\psi_e} \boldsymbol{v}_j]_{ik}(s, r)$.

**Expert wavelet integral blocks (EWIBs).** The EWIBs approximates the true operator by probabilistically combining the predictions of a set of local wavelet experts. While each local expert predicts a distinct feature, by mixing the individual predictions, the EWIB predicts the solution of a specific physical system. The global kernel $\mathcal{K}(\boldsymbol{a}(x); \phi)$ as a weighted combination of local wavelet experts weighted by the probabilities $\boldsymbol{\rho} \in \mathbb{R}^{d_e}$ is expressed as,

$$(\mathcal{K}(\phi \in \boldsymbol{\theta}) \boldsymbol{v}_j)(x) = \sum_{e=1}^{d_e} (\rho_e \cdot (\mathcal{K}_\phi^e \boldsymbol{v}_j)(x)); \quad x \in \Omega, \quad j \in h. \tag{1}$$

With $\sigma$ denoting the softmax function, the mixing probabilities $\boldsymbol{\rho} \in \mathbb{R}^{d_e}$ are estimated using context gate $\sigma(\mathcal{G}(e \mid \boldsymbol{v}(x), \tau; \boldsymbol{\theta}_\rho))$ on the input $\boldsymbol{v}(x)$ and equation label $\tau$. The context gate has its own model parameters $\boldsymbol{\theta}_\rho$. The final parameterization equation is expressed as,

$$(\mathcal{K}(\phi \in \boldsymbol{\theta}) \boldsymbol{v}_j)(x) = \sum_{e=1}^{d_e} \left( \rho_e \cdot \left\{ \mathcal{W}_{\psi_e}^{-1} \left( \sum_{i=1}^{d_v} (\mathcal{R}_\phi^e)_{ijk} [\mathcal{W}_{\psi_e} \boldsymbol{v}_j]_{ik}(s, r) \right)(x) \right\} \right). \tag{2}$$

The implementation steps of the EWIBs are provided in Algorithm 1.

**Dimension of the parameterization space.** For a $\mathbb{R}^d$ dimensional discretization and $d_a$ input features, we have $\boldsymbol{a} \in \mathbb{R}^{d \times d_a}$ and $\boldsymbol{v}_j \in \mathbb{R}^{d \times d_v}$. The DWT yields the detailed and approximation components $\mathbb{D}_v \in \mathbb{R}^{d_w \times d_v}$ and $\mathbb{A}_v \in \mathbb{R}^{d_w \times d_v}$, where $d_\omega = 2^{-s} d + 2(d_{\psi_e} - 1)$ with $d_\psi$ denoting the vanishing moment of the wavelet basis $\psi_e$. For a finite-dimensional parameterization space, we

choose to parameterize the kernels $\mathcal{R}_\phi^e$ in the highest wavelet compression level $s$. In the wavelet domain, we perform a global convolution by defining the kernels $\mathcal{R}_\phi^e \in \mathbb{R}^{d_\omega \times d_v \times d_v}$.

---

**Algorithm 1** Expert wavelet integral block (EWIB)

---

**Input:** $\boldsymbol{v}_j \in \mathbb{R}^{d \times d_v}$, PDE label $\tau$, and wavelet bases $\{\psi_e\}_{e=1}^{d_e}$.
1: Define the context gate: $\mathcal{G}(\varphi \in \boldsymbol{\theta}_\rho)$.
2: Obtain $\boldsymbol{\rho} \in \mathbb{R}^{d_e}$.
3: Define the local kernels: $\mathcal{R}_\phi^e \in \mathbb{R}^{d_\omega \times d_v \times d_v}$ for $e \in d_e$.
4: **for** $e \leftarrow d_e$ **do**
5:      $\{\mathbb{A}_v, \mathbb{D}_v\}_e \leftarrow [\mathcal{W}_{\psi_e} \boldsymbol{v}_j](s, r)(x) \in \mathbb{R}^{d_\omega \times d_v}$.                          ▷ Local DWT
6:      $\{\mathbb{A}_v^*, \mathbb{D}_v^*\}_e \leftarrow \sum_{i=1}^{d_v} (\mathcal{R}_\phi^e)_{(ijk)} (\{\mathbb{A}_v^*, \mathbb{D}_v^*\}_e)_{ik}$.                          ▷ Convolution
7:      $\boldsymbol{v}_{j+1}^e(x) \leftarrow \mathcal{W}_{\psi_e}^{-1} (\{\mathbb{A}_v^*, \mathbb{D}_v^*\}_e)$.                          ▷ Local IDWT
8: **end for**
9: Probabilisitic mixing of local prediction: $\boldsymbol{v}_{j+1} = \Gamma \left( \sum_{e=1}^{d_e} \rho_e \cdot \boldsymbol{v}_{j+1}^e + (g \cdot \boldsymbol{v}_j) \right)$.

---

**Context gates.**   Modeling complex and diverse heterogeneous tasks using multiple expert models is a well-studied topic in machine learning (Yuksel et al., 2012; Mattern, 2012). In m-PhOeNIX, combining predictions of individual local experts is facilitated by context gates $\mathcal{G}(v(x), \tau; \boldsymbol{\theta}_\rho)$, which map the equation label $\tau$ and the updated solution $v_j(x)$ from previous EWIBs (context information) to the local experts' probabilities $\boldsymbol{\rho} \in \mathbb{R}^{d_e}$ to be used for local mixing. Each EWIB $q_j$ has its own context gate $\mathcal{G}_j(v_j(x), \tau; \boldsymbol{\theta}_\rho)$. Given $\boldsymbol{v}_j \in \mathbb{R}^{d \times d_v}$, and $\tau \in \mathbb{Z}^+$, the probability vector $\boldsymbol{\rho}_j$ for $j^{th}$ EWIB is estimated as,

$$\rho_e = p(e \mid \boldsymbol{v}_j, \tau) = \frac{\exp\left( \mathcal{G}(e \mid \boldsymbol{v}_j, 1_{d_e}(\tau); \varphi \in \boldsymbol{\theta}_\rho) \right)}{\sum_{e=1}^{d_e} \exp\left[ \mathcal{G}(e \mid \boldsymbol{v}_j, 1_{d_e}(\tau); \varphi \in \boldsymbol{\theta}_\rho) \right]}, \ \rho_e \in \boldsymbol{\rho}_j, \tag{3}$$

where $1_{d_e}(\tau)$ is the one-hot encoding of the operator label $\tau$, and $\mathcal{G}(e \mid \boldsymbol{v}_j, 1_{d_e}(\tau); \boldsymbol{\theta}_p) : \mathbb{R}^d \mapsto \mathbb{R}^{d_e}$ is a context gate parameterized by $\boldsymbol{\theta}_\rho$, conditioned on the input $\boldsymbol{v}_j$ and the task label $\tau$.

**Multi-physics operator learning of heterogeneous physical systems.**   We consider $m$ set of physical systems defined by the differential operators $\mathcal{N}_\tau : \mathcal{A}_\tau \times \mathcal{U}_\tau \mapsto \mathcal{Q}_\tau$ for $\tau \in m$, where the pair $\{\mathcal{A}_\tau, \mathcal{U}_\tau, \mathcal{Q}_\tau\}$ denote the Banach spaces of input variables, solution, and source, which is allowed to differ across physical systems. Given the input and solution pairs $\{(\boldsymbol{a}_i^\tau, \boldsymbol{u}_i^\tau)_{i=1}^N\}_{\tau=1}^m$ and the operator labels: $\{\tau_j\}_{j=1}^m$, we approximate the multiphysics solution operator $\mathcal{D}_{\boldsymbol{\theta}}^{\mathcal{N}_{0:m}} : \mathcal{A}_{0:m} \mapsto \mathcal{U}_{0:m}$ using the m-PhOeNIX framework. The training involves simultaneous updates of EWIB and context gate parameters $\boldsymbol{\theta}$ and $\boldsymbol{\theta}_\rho$. During pre-training of the m-PhOeNIX model on initial $m$-PDEs, the order of the operators does not matter due to the label information $\tau \in \mathbb{Z}^+$ in the context gate.

With the pre-trained multi-physics operator $\mathcal{D}_{\boldsymbol{\theta}}^{\mathcal{N}_{0:m}} : \mathcal{A}_{0:m} \mapsto \mathcal{U}_{0:m}$, the sequential operator learning of new physical systems is done by fine-tuning the context gate $\mathcal{G}(\boldsymbol{\theta}_\rho)$. Since the backpropagation of loss through the EWIB parameters $\boldsymbol{\theta}$ is prohibited during sequential learning, the wall-clock time per epoch for fine-tuning reduces by a factor of half; for details, see Appendix B.3. Given a new system represented by the differential operator $\mathcal{N}_{m+1} : \mathcal{A}_{m+1} \mapsto \mathcal{U}_{m+1}$, and the training pairs $\{\boldsymbol{a}_i^{m+1}, \boldsymbol{u}_i^{m+1}\}_{i=1}^N$ the pre-trained m-PhOeNIX model is fine-tuned, where the loss function is only backpropagated to the context gate weights $\boldsymbol{\theta}\rho$.

Successful training results in the adapted solution operator $\mathcal{D}_{\boldsymbol{\theta}}^{\mathcal{N}_{0:m} \cup \mathcal{N}_{m+1}} : \{\mathcal{A}_{0:m} \cup \mathcal{A}_{m+1}\} \mapsto \{\mathcal{U}_{0:m} \cup \mathcal{U}_{m+1}\}$, which not only maps the inputs $\mathcal{A}_{m+1}$ of new system $\mathcal{N}_{m+1}$ to the solutions $\mathcal{U}_{m+1}$ but also predicts the solutions of previous seen differential operators $\{\mathcal{N}_0, \ldots, \mathcal{N}_m\}$ without catastrophic forgetting. This procedure is repeated for new operators $\tau \in \{\mathcal{N}_{m+1:M}\}$, where $M \gg m$. For every new system, the state dictionaries of the context gate parameters are stored locally, which is loaded along with the expert block parameter whenever inference on previous physical systems is required without catastrophic forgetting (Kirkpatrick et al., 2017) and rehearsal (Jeeveswaran et al., 2023). Thus, m-PhOeNIX eliminates the need to save large neural network models, thereby achieving data and resource efficiency. The schematic description of the multi-physics learning is portrayed in Figure 1, whereas the implementation steps are provided in Algorithm 2.

---

**Algorithm 2** Multi-physics operator learning of heterogeneous physical systems

---

**Input:** Training pairs $\{(\boldsymbol{a}_i^\tau, \boldsymbol{u}_i^\tau)_{i=1}^N\}_{\tau=1}^m$ for pre-training operators $\{\mathcal{N}_0, \ldots, \mathcal{N}_m\}$, PDE labels: $\{\tau_j\}_{j=1}^m$, wavelet bases $\{\psi_e\}_{e=1}^{d_e}$, and $\{(\boldsymbol{a}_i^\tau, \boldsymbol{u}_i^\tau)_{i=1}^N\}_{\tau=m+1}^M$ for new operators $\{\mathcal{N}_{m+1:M}\}$.

1: Pre-train m-PhOeNIX model: $\mathcal{D}_{\boldsymbol{\theta}}^{\mathcal{N}_{0:m}} : \mathcal{A}_\tau \mapsto \mathcal{U}_\tau$ for $\tau \in m$.
2: **for** new physical systems, $\tau \in \{\mathcal{N}_{m+1}, \ldots, \mathcal{N}_M\}$ **do**
3:     **for** epoch $\leftarrow$ epochs **do**
4:         Set gradient update 'False' for $\boldsymbol{\theta}$.
5:         Context information: $v_0^\tau(x) = \mathrm{P}(a^\tau(x))$.
6:         Predict from partially adapted model: $u_*^\tau(x) = \mathcal{D}_{\boldsymbol{\theta}}^{\mathcal{N}_{0:m} \cup \mathcal{N}_\tau^*}(\boldsymbol{a}^\tau)(x)$.
7:         Fine-tuning context gate: $\boldsymbol{\theta}_\rho \leftarrow \boldsymbol{\theta}_\rho - \alpha \nabla_{\boldsymbol{\theta}_\rho} \mathcal{L}\left(u^\tau(x), u_*^\tau(x)\right)$.
8:     **end for**
9:     **Output**: Adapted operator $\mathcal{D}_{\boldsymbol{\theta}}^{\mathcal{N}_{0:m} \cup \mathcal{N}_\tau} : \{\mathcal{A}_{0:m} \bigcup \mathcal{A}_\tau\} \mapsto \{\mathcal{U}_{0:m} \bigcup \mathcal{U}_\tau\}$.
10: **end for**

---

## 4 NUMERICAL ILLUSTRATIONS

We consider six 1D and six 2D time-dependent PDE examples, each representing a different physical system. Each example consists of 1400 training samples for different ICs, totaling 8400 1D and 2D training spatio-temporal trajectories. The performance in each example is examined for 100 test ICs, i.e., $6 \times 100 = 600$ different trajectories. Here, the physical systems refer to different governing PDEs, and the initial conditions are modeled as random fields using the Gaussian process (GP). The solution domain is considered as $\Omega \in [0, 1]$ with 257 spatial discretizations for 1D and $\Omega \in [0, 1]^2$ with $64 \times 64$ mesh for 2D illustrations. We have used 10 and 5 local wavelet experts in the 1D and 2D m-PhOeNIX models, respectively. Other details of model hyperparameters and compute resources are provided in detail in Appendix A.

**Problem setup.** We train four different multi-physics operator models. In (i) and (ii), we learn the multi-physics operators of 1D and 2D physical systems. The learned operators are denoted as $\mathcal{D}_{\boldsymbol{\theta}}^{\mathcal{N}_{0:m}} : u^{\{0:m\}}|_{\Omega \times [0,10]} \mapsto u^{\{0:m\}}|_{\Omega \times [11,T]}$, which maps the solutions $u^{\{0:m\}}|_{\Omega \times [0,10]}$ of $m$-physical systems at first 10-time steps in the domain $\Omega \in \mathbb{R}^d$ to the solutions at later time steps. This demonstration intends to showcase the capability of simultaneous operator learning of multiple heterogeneous physical systems without few-shot learning during prediction. In (iii) and (iv), we pre-train m-PhOeNIX models for 1D and 2D physical systems on initial two to three physical systems and then sequentially learn new solution operators of other heterogeneous physical systems. The pre-trained m-PhOeNIX models are adapted to new physical systems by sequentially fine-tuning only the context function. The sequentially adapted models are denoted as $\mathcal{D}_{\boldsymbol{\theta}}^{\mathcal{N}_{0:m} \cup \mathcal{N}_\tau} : u^{\{0:m\} \cup \mathcal{N}_\tau}|_{\Omega \times [0,10]} \mapsto u^{\{0:m\} \cup \mathcal{N}_\tau}|_{\Omega \times [11,T]}$, where $\mathcal{N}_{0:m} \cup \mathcal{N}_\tau$ represents the adaption to new physical system $\mathcal{N}_\tau$. This demonstration intends to display the capability of the adapted model to predict previously seen physical systems $\mathcal{N}_{0:m}$ in addition to the new system $\mathcal{N}_\tau$ without catastrophic forgetting.

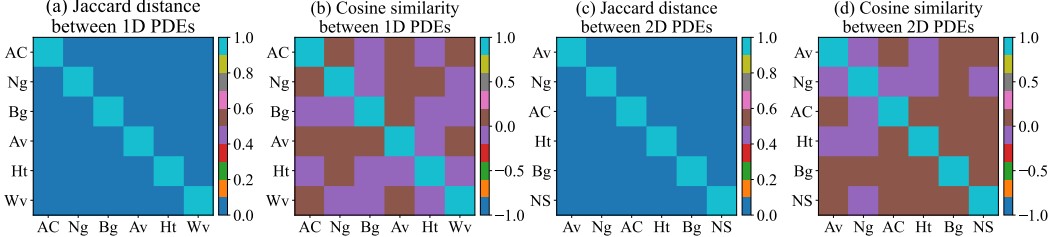

Figure 2: Similarity between the physical systems. Jaccard distance metric varies between 0 and 1, with 0 indicating no overlap and 1 complete overlap between PDEs. Cosine distance varies between -1 to 1, with 1 indicating a perfect match and 0 indicating a completely different PDE.

**Task Similarity.** Here, we demonstrate the heterogeneity between the undertaken tasks. In particular, we measure the similarity between the datasets of 1D and 2D physical systems using the Jaccard distance (Rajaraman & Ullman, 2011) and cosine similarity (Singhal et al., 2001). Examples of physical systems are Allen Cahn (Ac), Nagumo (Ng), Burgers (Bg), advection (Av), Heat (Ht), wave (Wv), and Navier-Stokes PDEs (NS). The data generation details are provided in Appendix F. The cosine similarity between the systems $\mathcal{N}_i$ and $\mathcal{N}_j$ is defined as $S_C(\mathcal{N}_i, \mathcal{N}_j) = (\mathcal{N}_i^T \mathcal{N}_j)/(\|\mathcal{N}_i\|\|\mathcal{N}_j\|)$, where $\|\cdot\|$ is the Frobenius Norm. The Jaccard distance is defined as $S_J(\mathcal{N}_i, \mathcal{N}_j) = 1 - |\mathcal{N}_i \cap \mathcal{N}_j|/|\mathcal{N}_i \cup \mathcal{N}_j|$. It is evident from the $S_C(\mathcal{N}_i, \mathcal{N}_j)$ and $S_J(\mathcal{N}_i, \mathcal{N}_j)$ metrics in Figure 2 that the operators are significantly different from each other.

**Simultaneous operator learning of multiple physical systems.** The proposed m-PhOeNIX presents an ensembling strategy between the integral kernels to support prediction on all the simultaneously trained heterogeneous physical systems without fine-tuning during inference. We train two m-PhOeNIX models on six 1D and 2D physical systems, each trained on 8400 different training samples from six different physical systems. The solutions at the first ten time steps are used to predict the solutions at the next 20 time steps for 1D and 10-time steps for 2D physical systems. The performance of trained models is tested by predicting solutions for a total of 600 initial conditions, 100 for each physical system. It is evident from the relative error in the temporal prediction in Figure 3 that the mean prediction error over the entire test dataset for each physical system is $< 2\%$ for 1D systems and $< 4\%$ in most of the cases for the 2D systems. This indicates that the m-PhOeNIX provides in-context operator learning of multiple heterogeneous physical systems.

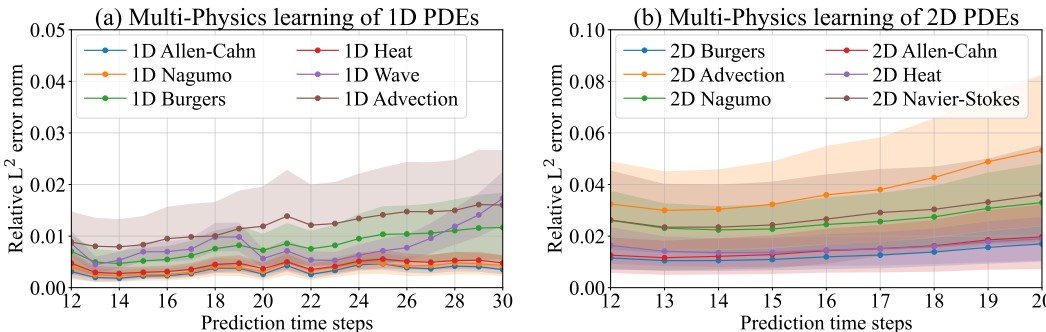

Figure 3: Prediction error of temporal evolutions of physical systems after simultaneously learning operators of all 8400 parametric PDEs. The shaded region indicates the 95% confidence interval (CI) of the prediction error over the test dataset involving 100 different ICs for each physical system.

**Sequential operator learning of heterogeneous physical systems.** We now investigate the efficacy of m-PhOeNIX against catastrophic forgetting of previously learned operators by sequentially learning solution operators of up to 4 for 1D and up to 3 for 2D heterogeneous physical systems. The pre-trained model is simultaneously trained on 2800 training samples from the Nagumo and Burgers equation for 1D systems and 4200 training samples from the Navier-Stokes, Allen-Cahn, and Burgers equation for 2D systems. The pre-trained models are then adapted to new physical systems by training the context gates sequentially on 580 samples from each new system. Sequentially learned models are tested on 100 test samples from previously seen and unseen future physical systems. The predictive accuracy of the temporal predictions on other tasks after being sequentially trained on another task is illustrated in Figure 4 and Figure 5. The prediction accuracy is estimated as $\upsilon = 1 - \varepsilon$, where $\varepsilon$ is the relative $L^2$ norm of the predictive error. The higher the value of the metric $\upsilon$, the better the predictions. It is evident in the results that the sequentially trained m-PhOeNIX models do not catastrophically forget previously seen physical systems.

**Zero-shot prediction on super-resolution.** Like existing task-specific neural operators, the m-PhOeNIX also exhibits discretization invariant properties without fine-tuning on a new resolution. While the multi-physics m-PhOeNIX is trained on a spatial resolution of 257 for 1D and $64 \times 64$ for 2D, we examine the zero-shot generalization to higher resolution by predicting the solutions at

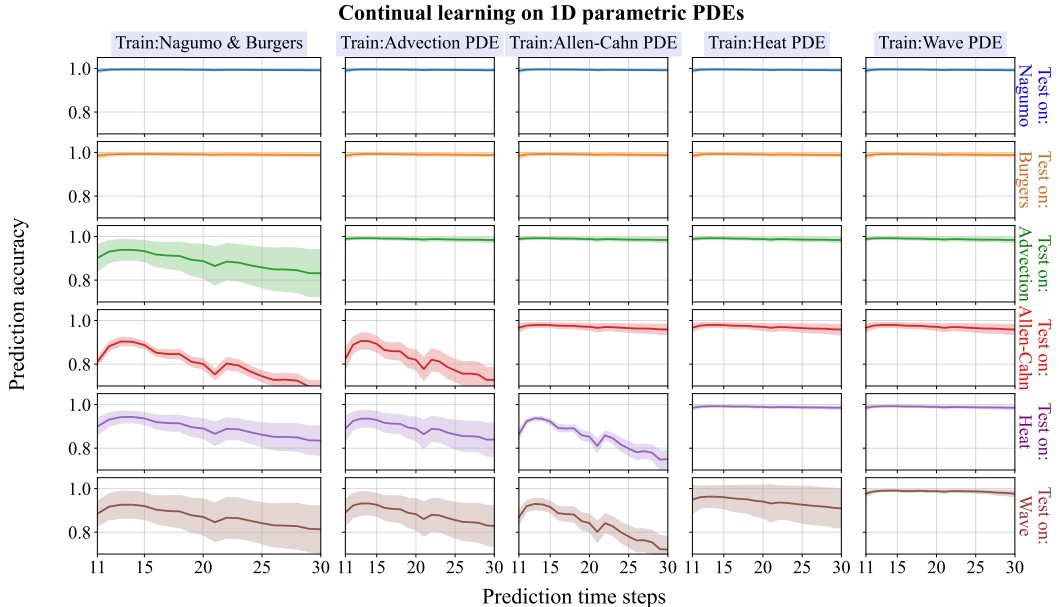

Figure 4: Sequential operator learning of 1D physical systems. The pre-trained m-PhOeNIX model is trained on Nagumo and Burgers equation, later sequentially trained on 4 new systems (indicated by columns), and sequentially tested on each system (indicated by rows). For e.g., consider the right column, which indicates the pre-trained m-PhOeNIX model's predictive performance on all systems after being sequentially trained on advection, Allen-Cahn, heat, and wave PDEs.

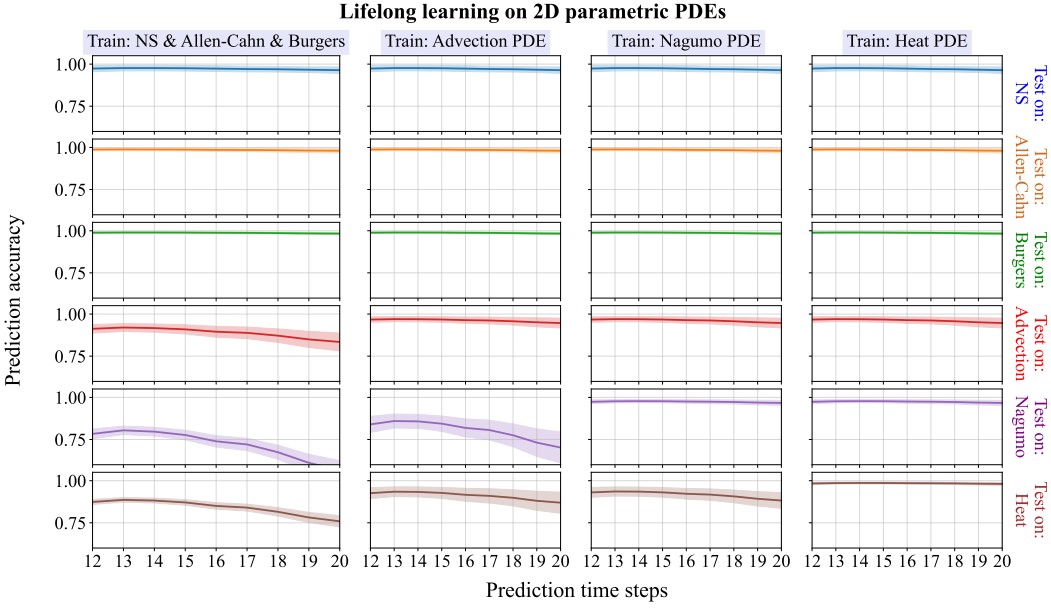

Figure 5: Sequential operator learning of 2D physical systems. An m-PhOeNIX model is pre-trained on Navier-Stokes, Allen-Cahn, and Burgers equations, later sequentially adapted to 3 new physical systems (indicated by columns), and sequentially tested on each task (indicated by rows).

resolutions of 514 for 1D and $128 \times 128$ for 2D examples. During zero-shot prediction, higher-resolution input fields are directly fed to the EWIBs. However, the context information is sub-sampled to training resolution before feeding to the context gates. The relative prediction error

averaged over the previous 100 different test samples is illustrated in Figure 6. The mean relative errors are found to increase to $< 6\%$ for 1D examples and, in most cases, $< 4\%$ for 2D examples.

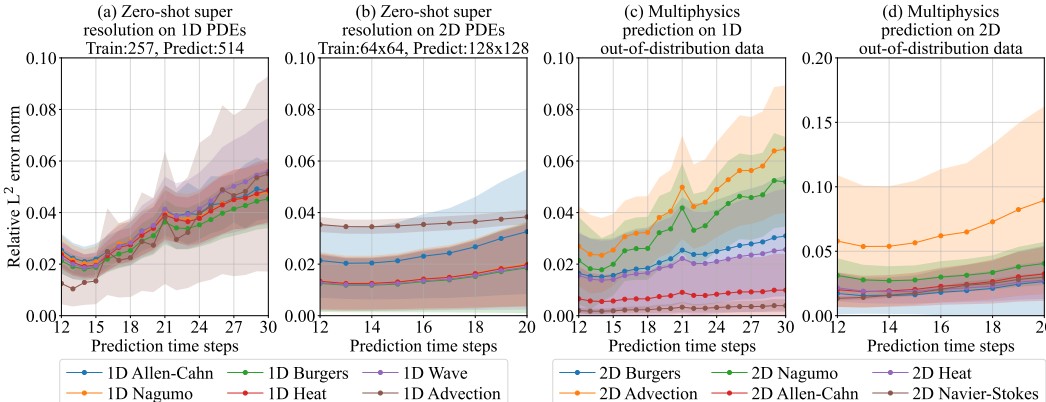

Figure 6: (a) & (b) Zero-shot prediction at higher resolution. The shaded region indicates the 95% CI of the prediction error, averaged over 100 different ICs at increased resolution. (c) & (d) Prediction error for out-of-distribution dataset. The shaded region indicates the 95% CI of the prediction error, averaged over 100 different out-of-distribution ICs.

**Out-of-distribution generalization.** Here, we examine the generalization ability of the pre-trained m-PhOeNIX models to initial random fields beyond the training distribution. The "out-of-distribution" datasets are generated using the same kernels as the in-distribution dataset but with different GP kernel parameters. The details are provided in Appendix F.2. The performance is assessed on 100 initial random fields from each physical system. The results in Figure 6 indicate that the relative error remains approximately $< 6\%$ for 1D and $< 5\%$ for most of the 2D problems.

Table 1: Performance against task-specific operators. m-PhOeNIX* indicates the multiphysics model trained on all six PDEs simultaneously. Overall best performing metric is indicated in bold blue color. Best performing multi-physics metric is indicated in bold brown color.

| Model | Performance on 1D PDEs (relative $L^2$ error norm in %) | | | | | |
|---|---|---|---|---|---|---|
| | Allen-Cahn | Nagumo | Burgers | Heat | Wave | Advection |
| DeepONet | 1.36±.56 | 2.29±.62 | 5.96±1.7 | 1.22±.46 | 1.25±.47 | 0.56±.42 |
| FNO | 0.28±.17 | **0.17**±.13 | 0.41±.32 | 0.25±.17 | 1.67±.52 | **0.15**±.09 |
| WNO | 0.66±.03 | 0.67±.31 | 2.21±2.0 | 0.79±.31 | 1.66±.49 | 0.35±.20 |
| CNO | 1.36±.63 | 0.94±.52 | 2.03±1.8 | 1.41±.59 | 1.47±.43 | 0.38±.20 |
| m-PhOeN. | **0.22**±.06 | 0.30±.16 | **0.31**±.15 | **0.11**±.06 | **0.34**±.09 | **0.15**±.09 |
| ICON | 7.86±.20 | 8.24±.21 | 9.09±.24 | 5.91±.22 | 9.93±.25 | 4.61±.17 |
| m-PhOeN.* | **0.37**±.09 | **0.41**±.18 | **0.88**±.42 | **0.47**±.16 | **1.01**±.27 | **0.44**±.21 |

| Model | Performance on 2D PDEs (relative $L^2$ error norm in %) | | | | | |
|---|---|---|---|---|---|---|
| | Navier-Stokes | Allen-Cahn | Burgers | Advection | Nagumo | Heat |
| DeepONet | 2.55±.45 | 0.83±.17 | 6.16±1.1 | 7.43±.67 | 1.76±.41 | 4.50±1.3 |
| FNO | 1.11±.28 | 0.22±.05 | 0.21±.07 | **0.17**±.02 | **0.21**±.05 | 1.23±.76 |
| WNO | **0.30**±.06 | 0.91±.18 | 0.48±.11 | 1.62±.17 | 0.92±.17 | 1.42±.82 |
| CNO | 2.87±1.8 | 0.86±.20 | 1.30±.55 | 0.93±.09 | 1.14±.32 | 1.22±.34 |
| m-PhOeN | 0.62±.13 | **0.19**±.04 | **0.14**±.03 | 0.38±.04 | 0.26±.05 | **0.34**±.19 |
| AVIT-B | 4.96±1.2 | 4.99±4.5 | 4.99±2.7 | 5.02±2.4 | 5.13±4.3 | 5.01±1.4 |
| AVIT-L | 4.98±0.5 | 5.01±0.4 | 5.01±0.4 | 5.01±0.1 | 4.98±0.1 | 5.01±0.2 |
| m-PhOeN.* | **2.58**±0.3 | **1.86**±1.1 | **2.17**±1.3 | **2.50**±1.6 | **3.10**±1.2 | **1.79**±0.4 |

**Comparison against existing multiphysics operators.** Here, we provide a comparison with the ICON (Yang et al., 2023) and MPP (McCabe et al., 2023) multi-physics operators. While the available ICON codes support only 1D problems, the available MPP codes are released for only 2D examples; thus, we have limited the comparison to 1D PDEs for ICON and to 2D PDEs for MPP. We simultaneously trained the ICON and MPP on six PDEs like the multiphysics m-PhOeNIX model. We have used the 31.56M parameter model of ICON from Yang et al. (2023) and AVIT-B (116M parameters) and AVIT-L (409M parameters) models of MPP from McCabe et al. (2023). On the contrary, the 1D and 2D m-PhOeNIX models have a size of 9.05M and 22.5M, i.e., the 1D m-PhOeNIX model is less than 1/3rd the size of ICON and the 2D m-PhOeNIX model is less than 1/5th the size of the MPP-AVIT-B and less than 1/16th the size of the MPP-AVIT-L model. The prediction errors are summarised in Table 1, where it is evident that even though the 1D and 2D m-PhOeNIX models are significantly smaller than the compared multi-physics models, m-PhOeNIX outperforms the compared models on the undertaken dataset.

**Problem-specific comparison.** We also compare the efficacy of the proposed m-PhOeNIX framework with existing problem-specific NOs like DeepONet, FNO, WNO, and CNO at the single-task level. For each system, we train independent problem-specific NOs. Alongside the multi-physics pre-trained m-PhOeNIX model, we also compare task-specific m-PhOeNIX models. The relative $L^2$ error norms of the predictions from different models averaged over 100 testing samples are provided in Table 1, where we observe that the m-PhOeNIX model, trained on a single task, performs outstandingly in most of the cases. Only in a few cases FNO beats the m-PhOeNIX results by a small margin. The multiphysics m-PhOeNIX model yields a relatively higher error as compared to the single-task m-PhOeNIX model. However, the accuracy of the multiphysics m-PhOeNIX model remains far better than the MPP models. These results also suggest that if required by fine-tuning the pre-trained multi-physics m-PhOeNIX model on a specific task, a high prediction accuracy can be achieved. We believe that by enlarging the model, the prediction error can further be minimized.

## 5 LIMITATIONS

The m-PhOeNIX framework extends the novel concept of distributed learning to NOs by introducing local wavelet experts as integral kernel approximators for multiple heterogeneous physical systems. However, in its current form, m-PhOeNIX requires a small initial trajectory to learn the time-dependent solution operators, which may be challenging to obtain for high-dimensional solution fields. Although initial studies in the appendix suggest that a high-dimensional model could potentially mitigate the need for an initial trajectory, detailed studies beyond the 1D case are not carried out. During zero-shot prediction on super-resolutions, the context gate currently takes sub-sampled inputs at the training resolution. To directly handle inputs at the super-resolution level, innovations such as an operator-enhanced context gate are needed, but these have not yet been introduced in this work. Additionally, this study does not address the implementation of the m-PhOeNIX framework on PDEs with irregular grids. Moreover, m-PhOeNIX requires system identities or labels as inputs, but how to differentiate between physical systems in an automated way during sequential learning, particularly when there is no clear task boundary, is not discussed.

## 6 DISCUSSIONS

We introduce the m-PhOeNIX framework for simultaneously and sequentially learning operators of multiple heterogeneous physical systems without catastrophic forgetting. While existing integral kernel-based NOs are effective at learning operators for a single physical system, m-PhOeNIX advances this by incorporating innovations such as modularity and local ensembling, enabling pre-training and sequential adaptation in scientific machine-learning tasks. The robustness of the framework is demonstrated through twelve 1D and 2D benchmark problems from computational mechanics. Additionally, we plan to enhance the shared feature space of the pre-trained m-PhOeNIX framework by utilizing large, diverse datasets from various physical systems, which will allow new operators to be sequentially learned with minimal samples. Our future efforts will also focus on integrating physics directly into the m-PhOeNIX model for data-free learning and simulating multi-physics phenomena.

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

## A    APPENDIX: DETAILS ON MODEL ARCHITECTURE

**Hyperparameters.**   Four expert wavelet integral blocks (WIBs) are used in the 1D and 2D m-PhOeNIX models. Each EWIB consists of 10 and 5 local wavelet experts for 1D and 2D models. The 1D EWIBs use orthogonal Daubechies wavelets (Daubechies, 1992) with vanishing moments from $1 \rightarrow 10$, and the 2D EWIBs use Biorthogonal wavelets with vanishing moments from $1 \rightarrow 5$. These wavelets project the inputs to wavelet space in the local experts so as to capture the global and local patterns in features. The wavelet compression level is fixed at 4. As activation operator mish activation function (Misra, 2019) is used. The transformations P, Q, and $g(\cdot)$ are modeled as $1 \times 1$ convolutions with $d_v = 64$ channels. The 1D context gate is designed as a 5-layered deep feed-forward network (FNN) with 256, 128, 64, 32, and 10 perceptrons in each layer. The 2D context gate consists of two CNN layers with 64 and 32 kernels, each kernel of a size 3, which follows the FNN in the 1D context gate. In total, 1D and 2D m-PhOeNIX models have approximately 9.05 and 22.5 million parameters. The ADAM optimizer is used with 0.001 as the initial learning rate and $10^{-6}$ as weight decay. A step scheduler with 20 stepsize and 0.5 decay rate is used. During the sequential learning of new physical systems, the decay step is modified to 15.

**Computational complexity of EWIBs.**   Given an input field $\boldsymbol{v}_j(x) \in \mathbb{R}^d$, the DWT and IDWT have $\mathcal{O}(d)$ time complexity. An $s$ level of wavelet compression results in wavelet coefficients of size $\mathbb{R}^{d_w}$. Spectral convolution of the coefficients incurs $\mathcal{O}(d_w)$ computational time. Designing $g(\cdot)$ as a $1 \times 1$ convolution incurs $\mathcal{O}(d)$ time. As $d_w < d$, the time complexity of an EWIB is $\mathcal{O}(dd_e)$.

**Compute resources.**   All training, fine-tuning, and testing are performed on a Ubuntu 20.04 system with a 12-core Xeon Silver 4214R Processor and a single Nvidia RTX A5000 24GB GPU card. The models are developed, trained, and fine-tuned in PyTorch 1.12.1. The wavelet decomposition is performed using the Pytorch Wavelets 1.3.0 (Cotter, 2020).

## B    APPENDIX: ABLATION STUDIES

### B.1    MULTI-PHYSICS LEARNING WITHOUT INITIAL TRAJECTORY

Here, we examine the ability of the m-PhOeNIX framework to learn the multi-physics solution operator of heterogeneous physical systems from single initial conditions instead of the initial trajectory. We carry out the study on the 1D physical systems, where we learn the multi-physics operator $\mathcal{D}_{\boldsymbol{\theta}}^{\mathcal{N}_{0:m}} : u^{\{0:m\}}|_{\Omega \times t_0} \mapsto u^{\{0:m\}}|_{\Omega \times [t_0, T)}$, which maps the initial conditions $u^{\{0:m\}}|_{\Omega \times t_0}$ of $m$-physical systems to the solutions at $t > t_0$. We train two multi-physics m-PhOeNIX models. The first model is the same 9.05 million parameter model used in the main results. In the second model, the uplifting channel dimension is increased to 100, resulting in a 20.88 million parameter model. The relative $L^2$ error norms of the predictions from these models are provided in Table 2. Given the same model size (9.05M), the relative errors are observed to be higher in the m-PhOeNIX model trained from only initial conditions. However, the relative error in the 20.88M model is found to be smaller than the 9.05M model, which indicates that a high-dimensional model is required for further improvement in the predictions. Overall, the results indicate that at the cost of a higher training cost, the m-PhOeNIX model can also effectively learn multi-physics solution operators from initial conditions without a need for the initial trajectory.

Table 2: Prediction error (%) of m-PhOeNIX models after training from initial conditions.

| Model size | Allen-Cahn | Nagumo | Burgers | Heat | Wave | Advection |
|---|---|---|---|---|---|---|
| 9.05M | 1.84±1.69 | 1.99±1.77 | 2.02±1.16 | 2.08±1.05 | 1.21±0.40 | 0.73±0.27 |
| 20.88M | 1.13±1.05 | 0.68±0.15 | 1.23±0.26 | 1.67±0.37 | 1.99±0.55 | 0.27±0.28 |

### B.2    EFFECT OF NUMBER OF LOCAL WAVELET EXPERTS

Here, we examine the effect of the number of local wavelet experts inside the EWIBs on the performance of the m-PhOeNIX framework. We conducted the study by sequentially learning the solution

operators of the 1D physical systems by considering 4, 7, 10, 13, and 16 local wavelet experts inside each EWIB. The number of EWIBs and other hyperparameters is kept the same as before. In all the cases, the pre-trained model is trained on the Nagumo and Burgers equation simultaneously and later adapted to Advection, Allen-Cahn, Heat, and Wave equations. The relative prediction error on a physical system after learning the corresponding physical systems is portrayed in Figure 7(a). It is evident in the results that increasing the number of experts in EWIBs not only decreases the relative prediction error over the pre-training physical systems but also prevents the relative error from increasing over the sequentially learned equations. However, it is necessary to note that increasing the number of experts increases the training time epoch, which according to our compute resources are found to be $\approx$ 57s, 75s, 105s, 173s, and 210s of computer wall time per epoch for 4, 7, 10, 13, and 16 experts, respectively.

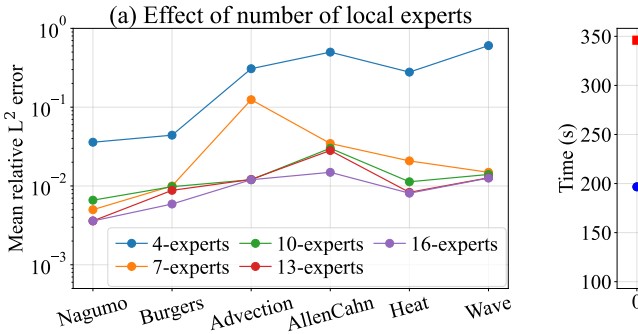 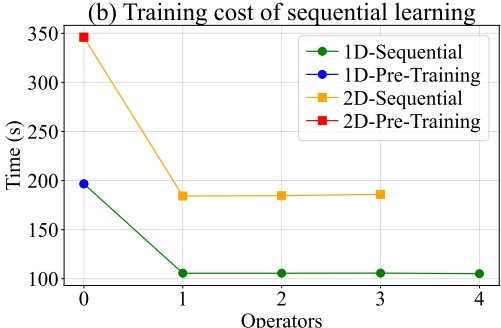

Figure 7: Summary of the ablation study. (a) Effect of the number of local wavelet experts on the performance of m-PhOeNIX over the considered 1D physical systems. (b) Computer wall time for one epoch for pre-training and sequential learning of new physical systems. The pre-trained model for 1D equations is trained on Nagumo and Burgers equations simultaneously, and for 2D equations is trained on Navier-Stokes, Allen-Cahn, and Burgers equations simultaneously.

### B.3 TRAINING COST OF PRE-TRAINING AND SEQUENTIAL LEARNING

The proposed multi-physics operator network extends the concept of pre-training and finetuning from natural language processing to scientific machine learning, where we train an initial model on two to three physical systems and then adapt to new PDEs through local ensembling. In particular, the pre-training is performed on 2800 training samples from 2 physical systems (1400 each) for 1D and 4200 training samples from 3 physical systems for 2D physical systems. The pre-trained model is later sequentially trained on 580 test samples from each new physical system. The training cost per epoch is illustrated in Figure 7(b). It is evident that with a pre-trained model, the cost of learning new equations is reduced by more than 2 times in both 1D and 2D tasks. Training the pre-trained models on more diverse physical systems and input conditions to enrich the feature space of m-PhOeNIX may yield a further reduction in training time during sequential learning.

## C APPENDIX: ARCHITECTURE DETAILS OF M-PHOENIX

**EWIBs.** The m-PhOeNIX models for both 1D and 2D examples contain 4 EWIBs, each containing 10 local wavelet experts. The local wavelet experts employ the Daubechies wavelets with the vanishing moments $1 \rightarrow 10$. The wavelet compression level is 4 across all the local wavelet experts. The "mish" activation function is used in all the EWIBs. The encoding transformation P is modeled as a $1 \times 1$ convolution with 64 kernels. The decoding transformation Q is modeled as a two-layer $1 \times 1$ convolution with 128 and 1 kernels, respectively. The linear skip transformation $g(\cdot)$ is also modeled as $1 \times 1$ convolution layer with dimensions of P, which in this case is 64 kernels. A summary is provided in Table 3.

**Optimizer.** The Adam optimizer with weight decay $10^{-3}$ is used. A step scheduler with step_size 20 and decay rate 0.5 is used for faster convergence. The loss is backpropagated over a batch size of

Table 3: Architecture details of multiphysics m-PhOeNIX models

|  | EWIBs | # experts | In-channel | Gates | Number of kernels in EWIBs | | | | # Parameters |
|---|---|---|---|---|---|---|---|---|---|
|  |  |  |  |  | (1) | (2) | (3) | (4) |  |
| 1D | 4 | 10 | 11 | 4 | 64 | 64 | 64 | 64 | 9.05M |
| 2D | 4 | 10 | 11 | 4 | 64 | 64 | 64 | 64 | 22.5M |

20. The network parameters are optimized for 150 epochs for 1D examples and 100 epochs for 2D examples.

**Context gate.** The context gate for 1D equations is modeled as a six-layered dense network with "mish" activation function in the hidden layers and "Softmax" at the output layer. The input size is $\mathbb{R}^d + 6$, where 6 denotes the number of classes in the one-hot encoding of task labels. The numbers of perceptions in the hidden layers are taken as $\{512, 256, 128, 64, 32\}$, with the output size as 10 for ten local wavelet experts. The context gate for 2D equations is modeled as three convolution networks followed by the three-layered dense network. The convolution networks are designed as Conv2D($c_{in}=c_{out}=64,k=5,s=2$), Conv2D($c_{in}=c_{out}=64,k=5,s=1$), and Conv2D($c_{in}=c_{out}=64,k=5,s=1$), where $k$ denotes kernel size and $s$ denotes stride. The dense network has the input size $256 + 6$, with the perceptions in the hidden layers as $\{128, 64, 10\}$. The setting of activation functions is kept the same as 1D.

# D APPENDIX: ARCHITECTURE DETAILS OF THE TASK-SPECIFIC OPERATORS

For the comparison against the existing task-specific operators like DeepONet (Lu et al., 2021), FNO (Li et al., 2020a), WNO (Tripura & Chakraborty, 2023), and CNO (Raonic et al., 2024), we train an independent NO for every different physical system. The same dataset used for training the simultaneous and sequential models is considered here. To highlight, the spatial resolutions are considered as 257 for 1D examples and $64 \times 64$ for the 2D examples. In all the examples, the aim is to approximate the operator $\mathcal{D} : u|_{\Omega \times [0,10]} \times \boldsymbol{\theta} \mapsto u|_{\Omega \times [11,T)}$, which maps the solutions at first 10-time steps in the domain $\Omega$ to the solutions at later time steps, in a similar manner to the main results presented in this study. The architectures of the compared models are as follows:

- **DeepONet**: For the 1D problems, we design the branch net as two-layer feed-forward networks (FFNs) with 512 perceptrons at each layer. The trunk net is designed as a three-layer FFN with 512 perceptions at each layer. The Adam optimizer is run for $2.5 \times 10^5$ iterations with a learning rate of $1 \times 10^{-3}$. The ReLU activation is used in the branch and trunk nets.

  For 2D problems, the branch net is designed with three convolution layers and 2 feed-forward layers. The convolution layers have 64, 128, and 128 kernels with the sizes 5, 5, and 3, respectively. A stride of 2 is used in the first two convolution layers. The feed-forward layers have 128 perceptions at each layer. The trunk net is designed as a four-layer FFN with 128 perceptions in each layer. Here, the Adam optimizer is run for $10^5$ iterations with an initial learning rate of $3 \times 10^{-4}$, which is reduced at every $2 \times 10^4$ iterations using a step decay rate of 0.5. The same ReLU activation is used in the branch and trunk nets.

- **FNO**: For both 1D and 2D examples, four Fourier blocks are used, with 64 in and out channels in each Fourier block. The Fourier modes are considered as 16 for 1D and 12 for 2D problems. The parameters are optimized using the Adam optimizer. A total of 500 epochs with 20 batch sizes are used in both 1D and 2D problems. An initial learning rate of $1 \times 10^{-3}$ is used, which is reduced at every 50 epoch for 1D and 100 epoch for 2D problems with the decay rate of 0.5. In all examples, the GeLU activation is used. A weight decay of $1 \times 10^{-4}$ is also utilized in the problems.

- **WNO**: Similar to FNO, four wavelet blocks are used in all the 1D and 2D examples. Each wavelet block has 64 and 32 kernels in 1D and 2D problems. The level of wavelet decomposition is considered as 5 for 1D and 4 for 2D problems. The parameters are optimized using the Adam optimizer. A total of 500 epochs with 20 batch sizes are used in the 1D

problems, whereas in 2D problems, a total of 200 epochs with 20 batch sizes are used. An initial learning rate of $1 \times 10^{-3}$ is used, which is reduced at every 50 epoch for 1D problems and 100 epoch for 2D problems at a rate of 0.5. In all examples, the GeLU activation and a weight decay of $1 \times 10^{-4}$ is used.

- **CNO**: For training the CNO framework on 1D problems, we use 4 numbers of downsampling block $\mathcal{D}$, 4 numbers of upsampling block $\mathcal{U}$, 4 numbers of invariant block $\mathcal{I}$, and 4 numbers of resnet block $\mathcal{R}$. For details on these blocks, see Raonic et al. (2024). The channel multiplier is set to 16. Except for 2D Advection and Allen-Cahn problems, we increase the number of resnet blocks to 6, whereas for 2D Advection and Allen-Cahn problems, we use 4 resnet blocks. For all 2D problems, the channel multiplier is set to 32. The convolution kernel size is taken as 3.

  The AdamW optimizer with a total of 500 epochs with 50 batch sizes is used in all the problems. An initial learning rate of $1 \times 10^{-3}$ is used, which is reduced at every 50 epoch at a rate of 0.5 for 1D problems and 10 epoch at a rate of 0.98 for 2D problems. The 'cno_lrelu' mentioned in the CNO paper is used as an activation function. The weight decay is set to $1 \times 10^{-6}$ for 1D and $1 \times 10^{-8}$ for 2D problems. The filter details are kept the same as those on the CNO paper.

- **m-PhOeNIX**: Four EWIBs are used in all the examples. In 1D problems, ten local wavelet experts are considered in each WIB, whereas in 2D problems, four local wavelet experts are considered. In 1D problems, the local wavelet experts are parameterized using Daubechies wavelets with vanishing moments $1 \rightarrow 10$, and in 2D problems, all the local wavelet experts are parameterized using bi-orthogonal wavelets. The level of wavelet decomposition is considered as 4 for all the problems. In the case of 1D problems, 32 kernels and in the case of 2D problems, 30 kernels are used in each local wavelet expert.

  Adam optimizer is used in all the problems. A total of 150 epochs with 20 batch sizes are used to optimize the parameters. The weight decay is set at $1 \times 10^{-6}$ in all problems. An initial learning rate of $1 \times 10^{-3}$ is used in the problems, which is reduced at every 25 epochs with a decay rate of 0.5. The local wavelet experts use mish activation to solve all the problems.

The size of model parameters is provided in Table 4.

Table 4: Details of the model parameters. m-PhOeNIX$^*$ indicates the multiphysics model trained on all six PDEs simultaneously.

| Model | Model parameters of 1D PDEs | | | | | |
| --- | --- | --- | --- | --- | --- | --- |
| | Allen-Cahn | Nagumo | Burgers | Advection | Heat | Wave |
| DeepONet | 922k | 922k | 922k | 922k | 922k | 922k |
| FNO | 551k | 551k | 551k | 551k | 551k | 551k |
| WNO | 877k | 746k | 615k | 877k | 615k | 615k |
| CNO | 672k | 672k | 672k | 672k | 672k | 672k |
| m-PhOeNIX | 2.87M | 2.87M | 2.87M | 2.87M | 2.87M | 2.87M |
| m-PhOeNIX$^*$ | $\longleftarrow$ 9.05M $\longrightarrow$ | | | | | |

| Model | Model parameters of 2D PDEs | | | | | |
| --- | --- | --- | --- | --- | --- | --- |
| | Advection | Nagumo | Allen-Cahn | Heat | Burgers | Navier-Stokes |
| DeepONet | 2.41M | 2.41M | 2.41M | 2.41M | 2.41M | 2.79M |
| FNO | 927k | 927k | 927k | 927k | 927k | 927k |
| WNO | 1.06M | 1.06M | 1.06M | 1.06M | 1.06M | 1.06M |
| CNO | 15.68M | 2.96M | 15.68M | 2.96M | 2.96M | 2.96M |
| m-PhOeNIX | 4.21M | 4.21M | 4.21M | 4.21M | 4.21M | 4.21M |
| m-PhOeNIX$^*$ | $\longleftarrow$ 22.5M $\longrightarrow$ | | | | | |

# E    APPENDIX: DETAILS OF MULTI-PHYSICS MODELS

## E.1    IN-CONTEXT OPERATOR NETWORKS (ICON) ARCHITECTURE

We have directly used the 31.56 million parameter ICON model from the paper by Yang et al. (2023) except for the batch size, which is modified to 16. The training is performed for 100K steps. The AdamW optimizer is used for optimizing the network parameters. For more details on the ICON-transformer architecture, see the supplementary material of Yang et al. (2023).

## E.2    MULTIPLE PHYSICS PRETRAINING (MPP) ARCHITECTURE

The AViT-B and AViT-L models of the MPP are directly used from the original paper by McCabe et al. (2023) with a change in the number of artificial epochs and number of time history. The artificial epoch size is kept at 400 so that for an epoch of 500, the total training step reaches 200K. The number of time histories to be used for prediction is kept at 10, the same as the training of m-PhOeNIX models. The AViT-B and AViT-L models have 116 million and 409 million parameters. For more details on the embedding dimension, dense layers, number of multi-heads, number of encoder-decoder blocks, and token size, please refer to McCabe et al. (2023).

# F    APPENDIX: DATA DESCRIPTION

## F.1    IN-DISTRIBUTION DATASET

The governing equations of motion of the example physical systems and the conditions used for generating training samples are provided in Table 5. Different conditions are simulated from Gaussian random fields (GRF) with radial basis function (RBF) kernel except for the Navier-Stokes equation. The RBF kernel is given as,

$$K(\boldsymbol{x}, \boldsymbol{x}\prime) = \sigma_k^2 \exp\left(\frac{-\|\boldsymbol{x} - \boldsymbol{x}\prime\|_2^2}{2\ell_k^2}\right), \qquad (4)$$

where the amplitude and lengthscale parameters $\sigma_k$ and $\ell_k$ for each of the physical problems are provided in Table 6. The domain of the examples is simulated given in Table 6. All the examples are simulated using periodic boundary conditions $u(\boldsymbol{x} = 0, t) = u(\boldsymbol{x} = 1, t)$. A total of 1400 training and 100 testing pairs are generated with different conditions for each physical system. While in multiphysics learning, all 1400 data are utilized, in sequential learning, only 580 pairs of data are used to learn new solution operators.

Table 5: Description of physical systems

| # | Physical system | Differential equation | Condition |
|---|---|---|---|
| 1 | Wave | $\partial_{tt}u(x,t) = \nu\Delta u(x,t)$ | $u(x,0)$ |
| 2 | Burgers | $\partial_t u(\boldsymbol{x},t) + 0.5\partial_x u^2(\boldsymbol{x},t) = \nu\partial_{xx}u(\boldsymbol{x},t)$ | $u(\boldsymbol{x},0)$ |
| 3 | Advection | $\partial_t u(\boldsymbol{x},t) + \alpha\partial_x u(\boldsymbol{x},t) = 0$ | $u(\boldsymbol{x},0)$ |
| 4 | Heat | $\partial_t u(\boldsymbol{x},t) = \alpha\Delta u(\boldsymbol{x},t)$ | $u(\boldsymbol{x},0)$ |
| 5 | Allen-Cahn | $\partial_t u(\boldsymbol{x},t) = \epsilon\partial_{xx}u(\boldsymbol{x},t) + u(\boldsymbol{x},t) - u(\boldsymbol{x},t)^3$ | $u(\boldsymbol{x},0)$ |
| 6 | Nagumo | $\partial_t u(\boldsymbol{x},t) = \nu\Delta u(\boldsymbol{x},t) + u(\boldsymbol{x},t)(1 - u(\boldsymbol{x},t))(u(\boldsymbol{x},t) - \alpha)$ | $u(\boldsymbol{x},0)$ |
| 7 | Navier-Stokes | $\partial_t \omega(\boldsymbol{x},t) + u(\boldsymbol{x},t) \cdot \nabla\omega(\boldsymbol{x},t) = \nu\Delta\omega(\boldsymbol{x},t) + f(\boldsymbol{x})$ 
 $\nabla \cdot u(\boldsymbol{x},t) = 0$ | $\omega(\boldsymbol{x},0)$ |

- For solving the 1D Burgers, 1D wave, 1D advection, and 1D heat equations, finite difference (FDM) codes are written. For solving the 1D Allen-Cahn and 1D Nagumo equation, codes are written using the pseudo-spectral element method. The time-forwarding of the solutions is done using a sampling frequency of 1000Hz. The space is discretized into 257 grid points in all the examples. The physical systems are solved using $\Delta t = 0.001$ seconds; however, the synthetic dataset is generated by recording the time-marching solutions at every $t$=0.2s, resulting in 50-time steps for each operator.

Table 6: Details of data generation. B.C. indicates the boundary condition.

| # | Physical system | Problem | Coefficients | Domain | Kernel parameters |
|---|---|---|---|---|---|
| 1 | Wave | 1D | $\nu = 0.1$ | $x \in [0,1]$, $t \in [0,10]$ | $\sigma = 0.1, \ell = 0.1$ |
| 2 | Burgers | 1D | $\nu = 10^{-3}$ | $x \in [0,1]$, $t \in [0,10]$ | $\sigma = 0.1, \ell = 0.1$ |
| | | 2D | $\nu = 10^{-3}$ | $x \in [0,1]^2$, $t \in [0,1]$ | $\sigma = 0.1, \ell = 0.25$ |
| 3 | Advection | 1D | $\alpha = 0.05$ | $x \in [0,1]$, $t \in [0,10]$ | $\sigma = 0.1, \ell = 0.25$ |
| | | 2D | $\alpha = 0.01$ | $x \in [0,1]^2$, $t \in [0,1]$ | $\sigma = 0.1, \ell = 0.3$ |
| 4 | Heat | 1D | $\alpha = 10^{-3}$ | $x \in [0,1]$, $t \in [0,10]$ | $\sigma = 0.1, \ell = 0.1$ |
| | | 2D | $\alpha = 10^{-3}$ | $x \in [0,1]^2$, $t \in [0,1]$ | $\sigma = 0.1, \ell = 0.25$ |
| 5 | Allen-Cahn | 1D | $\epsilon = 10^{-3}$ | $x \in [0,1]$, $t \in [0,10]$ | $\sigma = 0.1, \ell = 0.1$ |
| | | 2D | $\epsilon = 10^{-3}$ | $x \in [0,1]^2$, $t \in [0,1]$ | $\sigma = 0.1, \ell = 0.1$ |
| 6 | Nagumo | 1D | $\nu = 10^{-3}$ | $x \in [0,1]$, $t \in [0,10]$ | $\sigma = 0.1, \ell = 0.1$ |
| | | 2D | $\nu = 10^{-3}$ | $x \in [0,1]^2$, $t \in [0,1]$ | $\sigma = 0.1, \ell = 0.3$ |
| 7 | Navier-Stokes | 2D | $\nu = 10^{-3}$ | $x \in [0,1]^2$, $t \in [0,20]$ | – |

- The 2D Burgers and 2D advection equations are solved using the FDM method. The 2D heat, 2D Allen-Cahn, 2D Nagumo, and 2D Navier-Stokes equations are solved using the spectral element method. For solving 2D heat, 2D Allen-Cahn, and 2D Nagumo equations, a sampling frequency of 1000Hz is used. Similar to the 1D examples, the systems are solved using $\Delta t = 0.001$ seconds, while the synthetic dataset is created by recording the solutions at $\Delta t = 0.02$ seconds. For generating training data for the Navier-Stokes equation, the force field is generated as $f(x,y) = 0.1\left(\sin\left(2\pi\left(x+y\right)\right) + \cos\left(2\pi\left(x+y\right)\right)\right)$. The initial vorticity fields are generated from a GRF $\mathcal{N}(0, 7^{3/2}(-\Delta + 49\boldsymbol{I})^{-2.5})$. The time evolution of the solutions is predicted using the Crank–Nicolson scheme with a $\Delta t = 10^{-4}$s, whereas the data for training are recorded at every $t$=1s. For more details, see Li et al. (2020a).

## F.2 OUT-OF-DISTRIBUTION DATASET

To examine the robustness of the proposed multi-physics operator against out-of-distribution training operators, we generate the out-of-distribution testing dataset from different RBF kernel parameters. A total of 100 out-of-distribution samples for each equation are generated. The description of the kernel parameters is provided in Table 7. Note that other settings, such as the systems parameters and the domain, are kept the same as the training conditions.

Table 7: Out-of-distribution data generation details. $\mathcal{U}(\cdot, \cdot)$ denotes uniform distribution.

| 1D Equation | Coefficients | Domain | Kernel parameters |
|---|---|---|---|
| Allen-Cahn | $\epsilon = 10^{-3}$ | $x \in [0,1]$, $t \in [0,10]$ | $\sigma = \mathcal{U}(0.05, 0.5), \ell = \mathcal{U}(0.01, 0.5)$ |
| Nagumo | $\nu = 10^{-3}$ | $x \in [0,1]$, $t \in [0,10]$ | $\sigma = \mathcal{U}(0.05, 1), \ell = \mathcal{U}(0.01, 0.5)$ |
| Burgers | $\nu = 10^{-3}$ | $x \in [0,1]$, $t \in [0,10]$ | $\sigma = \mathcal{U}(0.05, 0.5), \ell = \mathcal{U}(0.01, 0.5)$ |
| Advection | $\alpha = 0.05$ | $x \in [0,1]$, $t \in [0,10]$ | $\sigma = \mathcal{U}(0.05, 1), \ell = \mathcal{U}(0.01, 0.5)$ |
| Heat | $\alpha = 10^{-3}$ | $x \in [0,1]$, $t \in [0,10]$ | $\sigma = \mathcal{U}(0.05, 0.25), \ell = \mathcal{U}(0.01, 0.5)$ |
| Wave | $\nu = 0.1$ | $x \in [0,1]$, $t \in [0,10]$ | $\sigma = \mathcal{U}(0.05, 0.5), \ell = \mathcal{U}(0.01, 0.4)$ |

| 2D Equation | Coefficients | Domain | Kernel parameters |
|---|---|---|---|
| Allen-Cahn | $\epsilon = 10^{-3}$ | $x \in [0,1]^2$, $t \in [0,1]$ | $\sigma = \mathcal{U}(0.05, 0.2), \ell = \mathcal{U}(0.1, 0.5)$ |
| Nagumo | $\nu = 10^{-3}$ | $x \in [0,1]^2$, $t \in [0,1]$ | $\sigma = \mathcal{U}(0.05, 0.2), \ell = \mathcal{U}(0.1, 0.5)$ |
| Burgers | $\nu = 10^{-3}$ | $x \in [0,1]^2$, $t \in [0,1]$ | $\sigma = \mathcal{U}(0.05, 0.2), \ell = \mathcal{U}(0.1, 0.5)$ |
| Advection | $\alpha = 0.01$ | $x \in [0,1]^2$, $t \in [0,1]$ | $\sigma = \mathcal{U}(0.05, 0.2), \ell = \mathcal{U}(0.1, 0.5)$ |
| Heat | $\alpha = 10^{-3}$ | $x \in [0,1]^2$, $t \in [0,1]$ | $\sigma = \mathcal{U}(0.05, 0.2), \ell = \mathcal{U}(0.1, 0.5)$ |
| Navier-Stokes | $\nu = 10^{-3}$ | $x \in [0,1]^2$, $t \in [0,20]$ | $\mathcal{N}(0, 2^3(-\Delta + 49\boldsymbol{I})^{-5})$ |

We demonstrate the degree of heterogeneity between the in- and out-of-distribution datasets in Figure 8. To demonstrate the heterogeneity, we use the Jaccard distance and cosine similarity as the heterogeneity measures. The cosine similarity between the systems $\mathcal{N}_i$ and $\mathcal{N}_j$ is defined as $S_C(\mathcal{N}_i, \mathcal{N}_j) = (\mathcal{N}_i^T \mathcal{N}_j)/(\|\mathcal{N}_i\| \|\mathcal{N}_j\|)$, where $\| \cdot \|$ is the Frobenius Norm. The Jaccard distance is defined as $S_J(\mathcal{N}_i, \mathcal{N}_j) = 1 - |\mathcal{N}_i \cap \mathcal{N}_j|/|\mathcal{N}_i \cup \mathcal{N}_j|$. It is evident that the out-of-distribution datasets are significantly different from the in-distribution datasets.

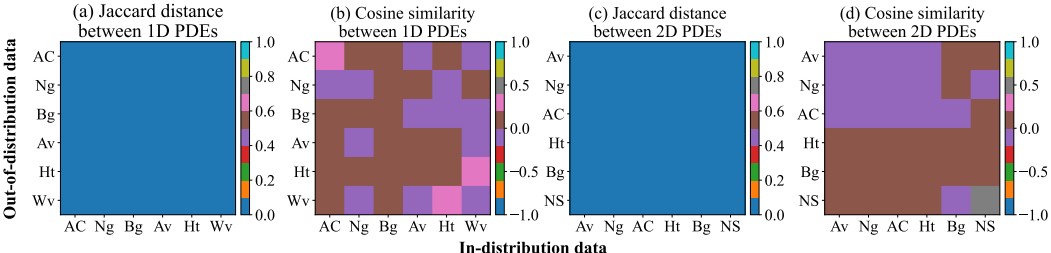

Figure 8: Similarity between the in- and out-of-distribution datasets. Jaccard distance metric varies between 0 and 1, with 0 indicating no overlap and 1 complete overlap between PDEs. Cosine distance varies between -1 to 1, with 1 indicating a perfect match and 0 indicating a completely different PDE.

## G    APPENDIX: SOLUTION TRAJECTORIES

In this section, we show the solution trajectory of a representative sample of the pre-trained and sequentially trained 2D physical systems. The m-PhOeNIX model was pre-trained on 1400 trajectories from Incompressible Navier-Stokes, Allen-Cahn, and Burgers' equation, and sequentially adapted on the Advection, Nagumo, and Heat equation using 580 trajectories from each system.

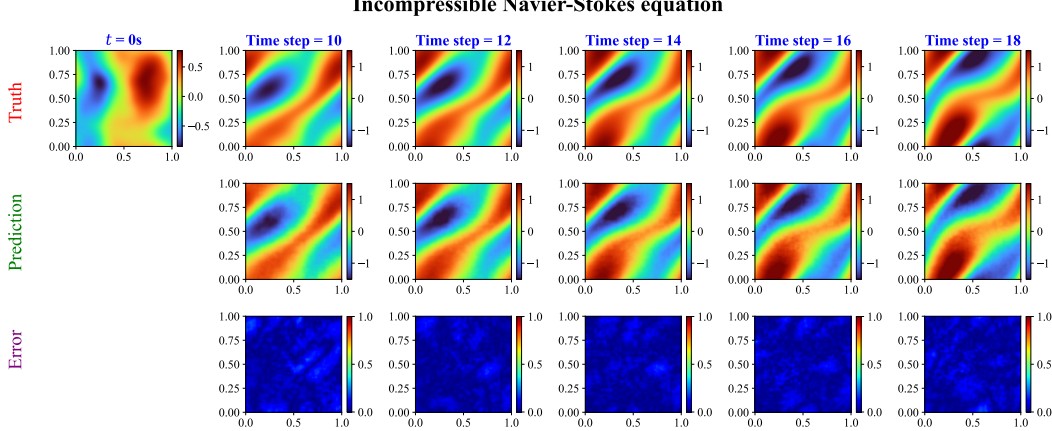

Figure 9: Solutions trajectories of pre-trained incompressible Navier-Stokes equation. The solution trajectory is shown for a representative initial condition.

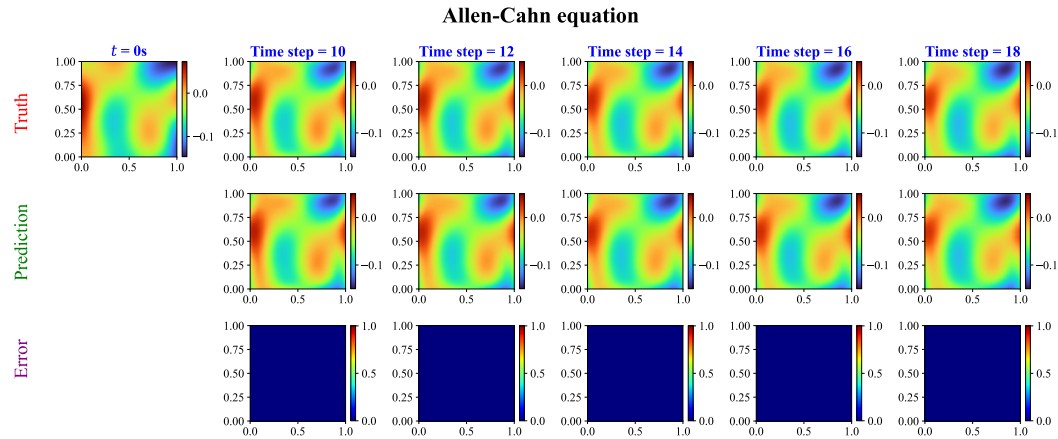

Figure 10: Solutions trajectories of pre-trained Allen-Cahn equation. The solution trajectory is shown for a representative initial condition.

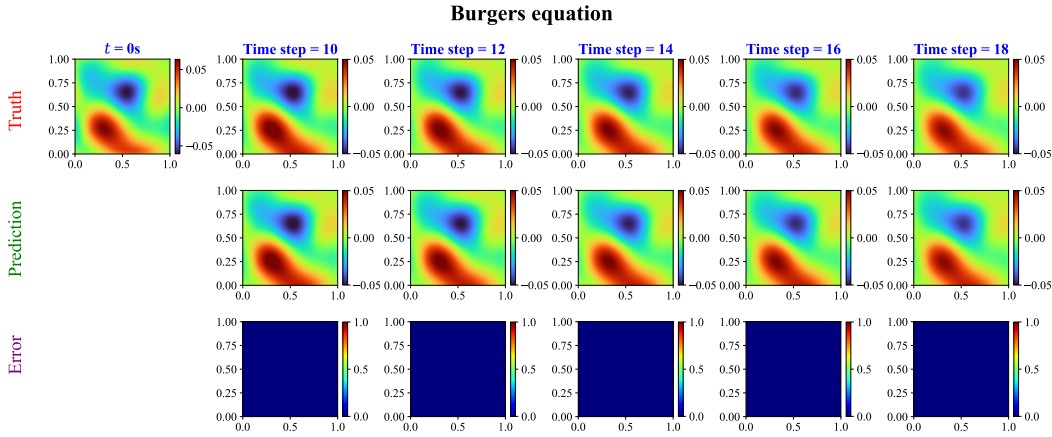

Figure 11: Solutions trajectories of pre-trained Burgers' equation. The solution trajectory is shown for a representative initial condition.

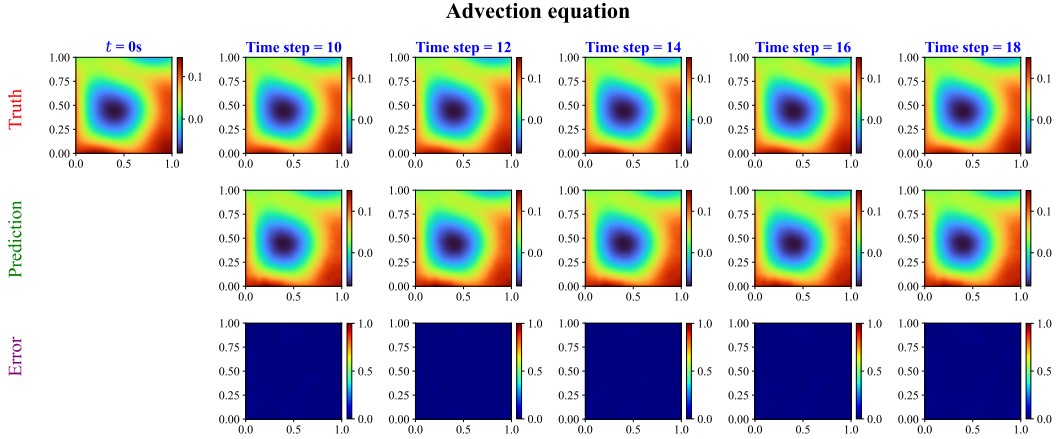

Figure 12: Solutions trajectories of sequentially trained Advection equation. The solution trajectory is shown for a representative initial condition.

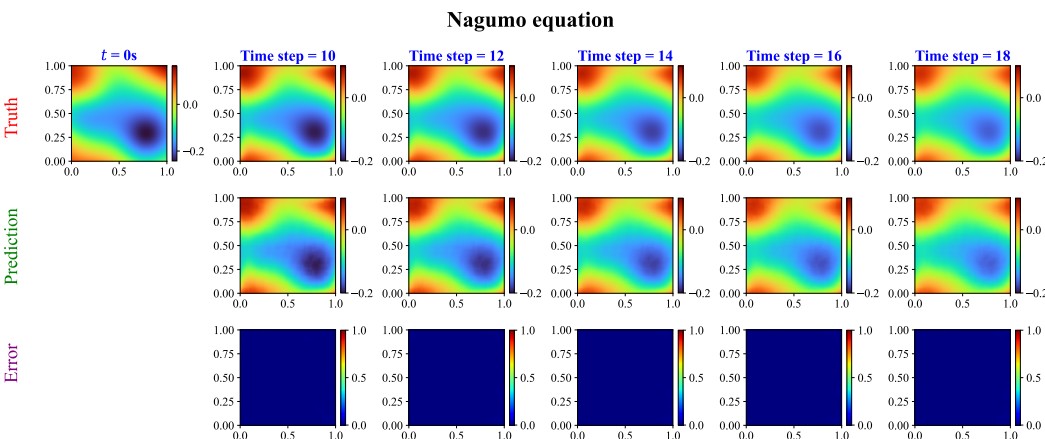

Figure 13: Solutions trajectories of sequentially trained Nagumo equation. The solution trajectory is shown for a representative initial condition.

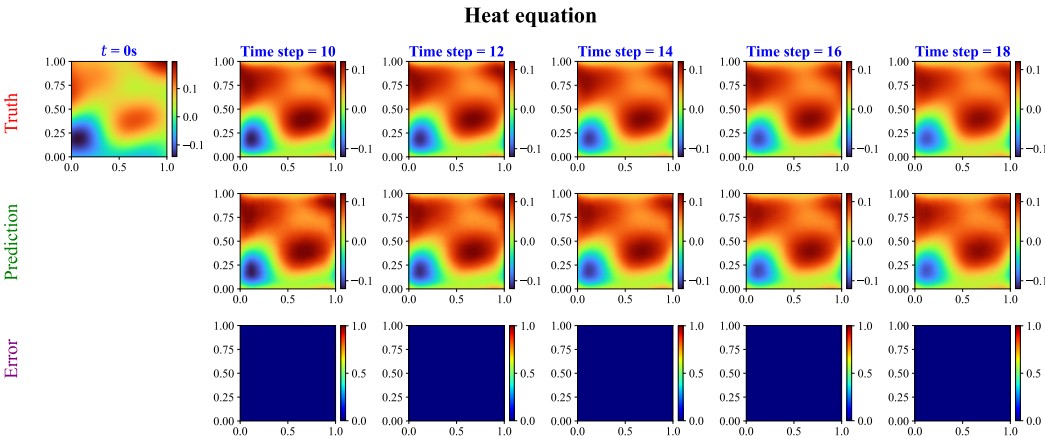

Figure 14: Solutions trajectories of sequentially trained Heat equation. The solution trajectory is shown for a representative initial condition.

## H   APPENDIX: OUTPUT PROBABILITIES CONTEXT GATES

Here, we illustrate the ensembling probabilities of the local wavelet experts. There are four hidden expert wavelet integral layers and, correspondingly, four context gates. Each hidden layer has ten local wavelet experts. The ensembling probabilities of the local experts predicted by the context gates are provided in Fig. 15. The probabilities are obtained by averaging over all the testing samples and time steps.

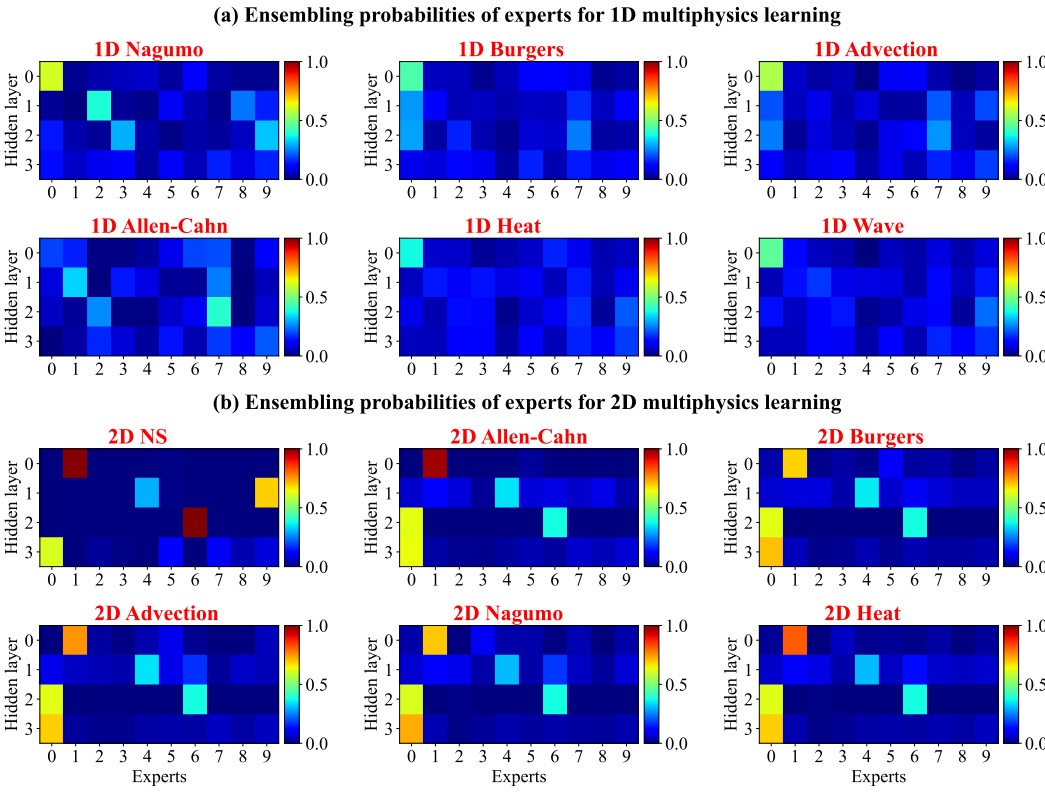

Figure 15: Probabilities resulting from context gates for different problems. Both 1D and 2D multi-physics models have 4 hidden layers, and each hidden layer is accompanied by its own context gate. Each context gate predicts the probabilities of ten local wavelet experts.

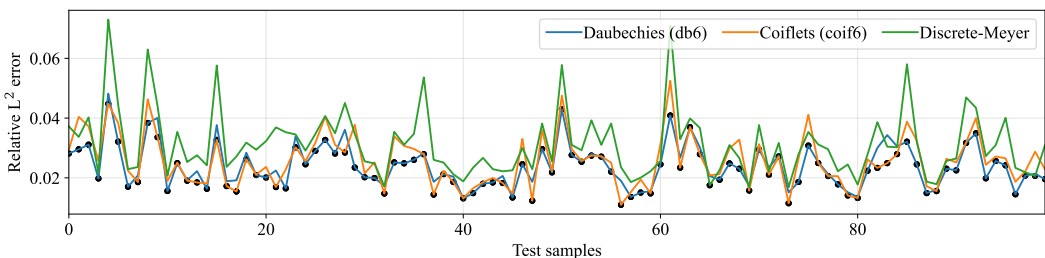

Figure 16: Relative L$^2$ error over 100 test samples of Darcy equation. The aim is to map permeability fields to pressure fields. The black dots denote the minimum error among the wavelet basses.

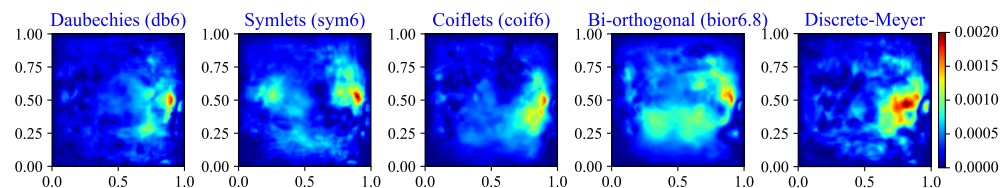

Figure 17: Distribution of absolute error over the spatial domain for a representative sample of Darcy equation in a rectangular domain. The aim is to map permeability fields to pressure fields.

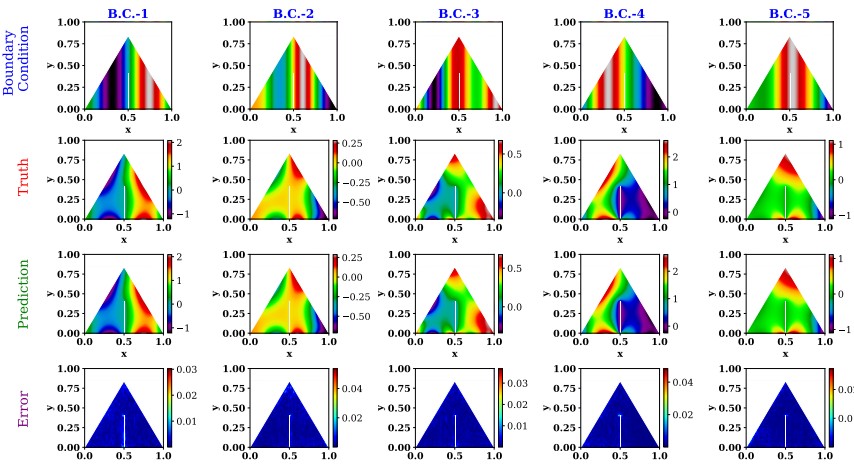

Figure 18: Performance of m-PhOeNIX to changing boundary conditions, exemplified on the Darcy equation in a triangular domain. The aim is to predict pressure fields for given boundary conditions.

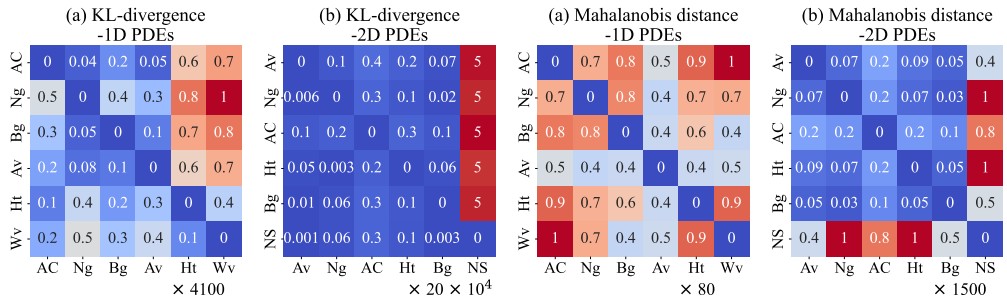

Figure 19: Kullback-Leibler (KL) divergence and Mahalanobis distance between the PDE datasets.

